# Integrating unsupervised language model with triplet neural networks for protein gene ontology prediction

Yi-Heng Zhu[1,2], Chengxin Zhang[2], Dong-Jun Yu[1]*, Yang Zhang[2,3]*

**1** School of Computer Science and Engineering, Nanjing University of Science and Technology, Nanjing, People's Republic of China, **2** Department of Computational Medicine and Bioinformatics, University of Michigan, Ann Arbor, Michigan, United States of America, **3** Department of Biological Chemistry, University of Michigan, Ann Arbor, Michigan, United States of America

* njyudj@njust.edu.cn (D-JY); zhang@zhanggroup.org (YZ)

**Data Availability Statement:** The benchmark dataset, standalone package, and online server for ATGO are available at https://zhanggroup.org/ATGO/.

## Abstract

Accurate identification of protein function is critical to elucidate life mechanisms and design new drugs. We proposed a novel deep-learning method, ATGO, to predict Gene Ontology (GO) attributes of proteins through a triplet neural-network architecture embedded with pre-trained language models from protein sequences. The method was systematically tested on 1068 non-redundant benchmarking proteins and 3328 targets from the third Critical Assessment of Protein Function Annotation (CAFA) challenge. Experimental results showed that ATGO achieved a significant increase of the GO prediction accuracy compared to the state-of-the-art approaches in all aspects of molecular function, biological process, and cellular component. Detailed data analyses showed that the major advantage of ATGO lies in the utilization of pre-trained transformer language models which can extract discriminative functional pattern from the feature embeddings. Meanwhile, the proposed triplet network helps enhance the association of functional similarity with feature similarity in the sequence embedding space. In addition, it was found that the combination of the network scores with the complementary homology-based inferences could further improve the accuracy of the predicted models. These results demonstrated a new avenue for high-accuracy deep-learning function prediction that is applicable to large-scale protein function annotations from sequence alone.

## Author summary

In the post-genome sequencing era, a major challenge in computational molecular biology is to annotate the biological functions of all genes and gene products, which have been classified, in the context of the widely used Gene Ontology (GO), into three aspects of molecular function, biological process, and cellular component. In this work, we proposed a new open-source deep-learning architecture, ATGO, to deduce GO terms of proteins from the primary amino acid sequence, through the integration of the triplet neural-network with pre-trained language models of protein sequences. Large benchmark tests

**Funding:** This work is supported in part by the China Scholarship Council (201906840041 to YHZ), the National Natural Science Foundation of China (62072243 and 61772273 to DJY), the Natural Science Foundation of Jiangsu (BK20201304 to DJY), the Foundation of National Defense Key Laboratory of Science and Technology (JZX7Y202001SY000901 to DJY), the National Institute of General Medical Sciences (GM136422, S10OD026825 to YZ), the National Institute of Allergy and Infectious Diseases (AI134678 to YZ), and the National Science Foundation (IIS1901191, DBI2030790, MTM2025426 to YZ). The funders had no role in study design, data collection and analysis, decision to publish, or preparation of the manuscript.

**Competing interests:** The authors have declared that no competing interests exist.

showed that, when powered with transformer embeddings of the language model, ATGO achieved a significantly improved performance than other state-of-the-art approaches in all the GO aspect predictions. Following the rapid progress of self-attention neural network techniques, which have demonstrated remarkable impacts on natural language processing and multi-sensory data process, and most recently on protein structure prediction, this study showed the significant potential of attention transformer language models on protein function annotations.

This is a *PLOS Computational Biology* Methods paper.

## Introduction

Proteins are the material basis of life and play many important roles in living organisms, such as catalyzing biochemical reactions, transmitting signals, and maintaining structure of cells [1]. To elucidate life mechanisms, it is critical to identify the biological functions of proteins, which have been grouped, in the context of the widely used Gene Ontology (GO), into three aspects of molecular function (MF), biological process (BP), and cellular component (CC) [2]. Direct determination of protein functions via biochemical experiments is standard but often time-consuming and incomplete [3]. As a result, numerous sequenced proteins have no available function annotation to date. As of June 2022, for example, the UniProt database [4] harbored ~230 million protein sequences, but only <1% of them were annotated with known GO terms using experimental evidence. To fill the gap between sequence and function, it is urgent to develop efficient computational methods for protein function prediction [5,6].

Existing function prediction methods use at least one of three information sources: template detection [7], biological network [8], and sequence composition [9]. Conventional function prediction methods typically rely on the detection of templates that have similar sequence or structure to the query and therefore suitable for functional inference. For examples, GOtcha [10], GoFDR [11], and Blast2GO [7] identify sequence templates using BLAST or PSI-BLAST alignments [12], while COFACTOR [13] and ProFunc [14] search templates through structure alignment [15]. Meanwhile, biological networks established by either protein-protein interaction (PPI) or gene co-expression have been used in more recent function predictors, such as NetGO [8], MS-kNN [16], and TripletGO [17]. The rationale behind PPI networks or gene co-expression is that the proteins with PPI or similar gene expression patterns are more likely to be involved in the same biological pathway or found in the same subcellular location.

Both template-based and network-based approaches have a common drawback: the accuracy of these methods is contingent upon the availability of readily identifiable and functionally annotated templates or interaction partners. To eliminate such dependence, machine learning-based methods have emerged to directly derive functions from the sequence composition of the query alone [8]. This can be achieved by extracting hand-crafted sequence features (e.g., k-mer amino acid coding and matches of protein domain families), which can then be used by machine learning approaches (e.g., support vector machine and logistic regression) to implement function model training and prediction, with typical examples including FFPred [18] and GOlabeler [19].

Despite the potential advantage, the prediction accuracy of many early machine learning methods was not satisfactory. One of the major reasons is due to the lack of informative feature

representation methods, as most of the approaches are based on simple feature representations, such as amino acid composition, physiochemical properties, and protein family coding, which cannot fully extract the complex pattern of protein functions [19,20]. To partly overcome this barrier, several methods, e.g., DeepGO [9], DeepGOPlus [21], and TALE [22], utilized deep learning technology to predict protein function. Compared to traditional machine learning approaches, one advantage of deep learning technologies is that they could extract more discriminative feature embeddings from preliminary sequence through designing complex neural networks. Nevertheless, the performance of deep learning methods is often hampered by the limitation and imbalance of annotated function data. Currently, there are only ~130,000 proteins with experimental annotations in the UniProt database, where 81.6% of GO terms are annotated to less than 50 proteins per term. The insufficient experimental data significantly limits the effectiveness of training the deep neural network models.

To alleviate the issue caused by the lack of annotated data, a promising approach is to utilize protein language models pre-trained through deep-learning networks on large-scale sequence databases which may not have functional annotations. Due to the extensive sequence training and learning, important inter-residue correlation patterns, which are critical for protein functions, can be extracted through the language models and utilized for functional embedding. Several protein language models, such as ProtTrans [23], models used in TAPE [24], and Bepler & Berger's approach [25], have been recently proposed in the literatures. Especially, an unsupervised protein language model, ESM-1b transformer [26], which utilized self-attention networks to learn the evolutionary diversity from 250 millions of protein sequences, has found impressive usefulness in protein contact-map and structure prediction. Meanwhile, the unsupervised language models have also been used, often through supervised learners such as convolutional neural networks (CNNs), for protein function prediction tasks, with examples including the predictions of protein molecular function [27], mutation and stability [24], subcellular localization [23], GO transferals [28], and ligand binding [29].

In this work, we proposed a new deep learning model, ATGO, for high accuracy protein function prediction by the integration of the triplet neural-network protocol [30] with the language models of protein sequences. First, we utilized an unsupervised self-attention transformer model, ESM-1b transformer [26], which has been pre-trained on millions of unannotated sequences, as a feature extractor to generate feature embeddings. Next, a supervised triplet neural-network was extended to train function annotation models from multi-layer transformer feature embeddings, by enhancing the difference between positive and negative samples. To improve prediction accuracy, we further implemented a composite version, ATGO+, by combining ATGO with a sequence homology-based model. Both ATGO and ATGO+ have been systematically tested on a large set of non-redundant proteins, where the results demonstrated significant advantage on accurate GO term prediction over the current state-of-the-art of the field. The standalone package and an online server of ATGO are made freely available through URL https://zhanggroup.org/ATGO/.

## Results

### Overall performance of ATGO

ATGO is a deep learning-based approach to protein function annotation with respect to GO terms. As shown in Fig 1, ATGO first extracts three layers of feature embeddings from the attention network-based transformer (ESM-1b), which are then fused by a fully connected neural network. Next, the fused feature embedding is trained by a triplet neural network to generate confidence scores of the predicted GO terms (See 'Methods and materials").

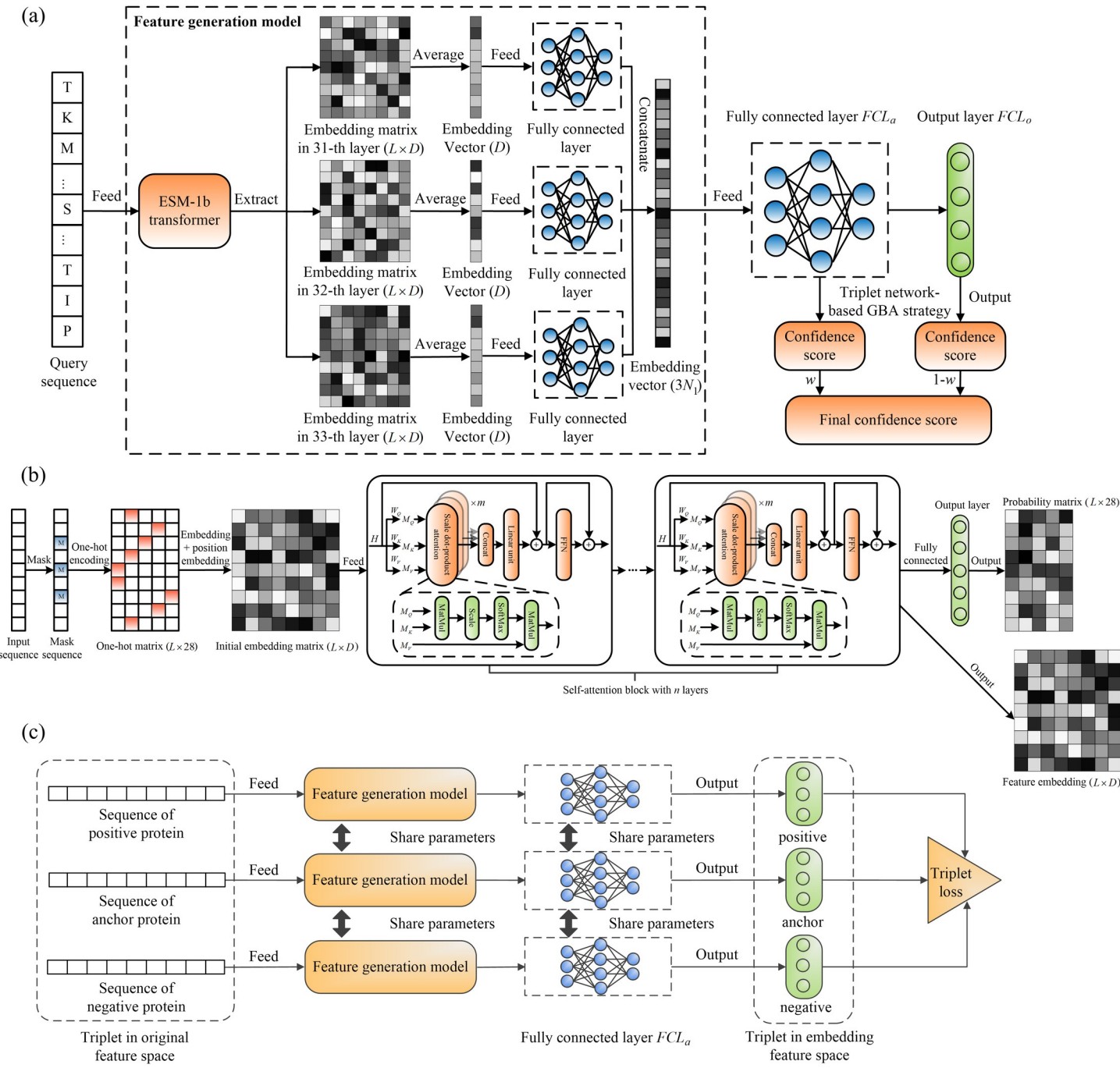

**Fig 1. The procedures of ATGO for protein function prediction.** **(a)** The workflow of ATGO. Starting from the input sequence, the ESM-1b transformer is utilized to generate the feature embeddings from the last three layers, which are fused by a fully connected neural network. The fused feature embedding is then fed into a triplet network to create confidence scores of GO models. **(b)** The structure of ESM-1b transformer. For an input sequence, the masking, one-hot encoding, and position embedding are orderly executed to generate the coding matrix, which is then fed into a self-attention block with $n$ layers. Each layer can output a feature embedding matrix from an individual evolutionary view through integrating $m$ attention heads with a feed-forward network, where the scale dot-product attention is performed in each head. **(c)** The design of a triplet network for assessing feature similarity. The input is a triplet variable ($anc$, $pos$, $neg$), where $anc$ is an anchor (baseline) protein, and $pos$ (or $neg$) is a positive (or negative) protein with the same (or different) function of $anc$. Each sequence is fed into the designed feature generation model to extract a feature embedding vector, as the input of fully connected layer to output a new embedding vector. Then, the feature dissimilarity between two proteins is measured by Euclidean distance of embedding vectors. Finally, a triplet loss is designed to enhance the relationship between functional similarity and feature similarity in embedding space.

**Table 1. The summary of the proposed ATGO/ATGO+ and other 10 competing GO prediction methods on the 1068 benchmark proteins.** *p*-values in parenthesis are calculated between ATGO and other single-based methods and between ATGO+ and other composite methods by two-sided Student's t-test. Specifically, the proposed ATGO and ATGO+ are repeatedly implemented with 10 times on the benchmark dataset to generate the corresponding performance evaluation indices, which are compared with the fixed evaluation index generated by the competing method to calculate *p*-value using two-sided Student's t-test. Bold fonts highlight the best performer in each category. Coverage is the ratio of the number of proteins with available prediction scores divided by the total number of test proteins.

| Methods | | $F_{max}$ | | | AUPR | | | Coverage | | |
|---|---|---|---|---|---|---|---|---|---|---|
| | | **MF** | **BP** | **CC** | **MF** | **BP** | **CC** | **MF** | **BP** | **CC** |
| Single algorithms | SAGP | 0.597 (1.1e-07) | 0.400 (5.5e-10) | 0.534 (1.5e-14) | 0.351 (3.1e-17) | 0.242 (4.8e-17) | 0.322 (1.6e-19) | 0.88 | 0.87 | 0.85 |
| | PPIGP | 0.224 (1.2e-18) | 0.303 (3.0e-17) | 0.467 (7.4e-17) | 0.103 (4.6e-20) | 0.181 (8.9e-19) | 0.340 (2.9e-19) | 0.52 | 0.63 | 0.63 |
| | NGP | 0.224 (1.2e-18) | 0.254 (1.2e-18) | 0.481 (1.7e-16) | 0.103 (4.6e-20) | 0.151 (2.1e-19) | 0.355 (5.0e-19) | 1.00 | 1.00 | 1.00 |
| | DeepGO | 0.355 (4.3e-17) | 0.317 (9.6e-17) | 0.499 (6.4e-16) | 0.293 (4.3e-18) | 0.218 (8.1e-18) | 0.430 (1.5e-17) | 1.00 | 1.00 | 1.00 |
| | FunFams | 0.476 (1.0e-14) | 0.315 (7.6e-17) | 0.424 (7.5e-18) | 0.294 (4.4e-18) | 0.152 (2.1e-19) | 0.236 (1.3e-20) | 0.66 | 0.62 | 0.58 |
| | DeepGOCNN | 0.328 (1.8e-17) | 0.307 (3.8e-17) | 0.463 (5.6e-17) | 0.264 (1.8e-18) | 0.208 (4.2e-18) | 0.337 (2.6e-19) | 1.00 | 1.00 | 1.00 |
| | DIAMONDScore | 0.592 (2.4e-08) | 0.391 (1.6e-11) | 0.511 (1.6e-15) | 0.272 (2.3e-18) | 0.209 (4.5e-18) | 0.239 (1.4e-20) | 0.80 | 0.81 | 0.78 |
| | TALE | 0.393 (1.8e-16) | 0.315 (7.7e-17) | 0.516 (2.7e-15) | 0.344 (2.4e-17) | 0.236 (3.0e-17) | 0.496 (1.6e-15) | 1.00 | 1.00 | 1.00 |
| | ATGO | **0.627** | **0.425** | **0.623** | **0.603** | **0.361** | **0.600** | 1.00 | 1.00 | 1.00 |
| Composite algorithms | DeepGOPlus | 0.603 (3.4e-10) | 0.409 (3.7e-11) | 0.533 (6.8e-17) | 0.528 (8.7e-14) | 0.323 (2.2e-15) | 0.486 (8.8e-18) | 1.00 | 1.00 | 1.00 |
| | TALE+ | 0.602 (3.3e-10) | 0.420 (8.4e-09) | 0.586 (2.2e-13) | 0.542 (5.6e-13) | 0.332 (2.2e-14) | 0.569 (3.5e-12) | 1.00 | 1.00 | 1.00 |
| | ATGO+ | **0.631** | **0.438** | **0.624** | **0.611** | **0.368** | **0.600** | 1.00 | 1.00 | 1.00 |

To examine the efficacy of proposed ATGO, we implemented three GO prediction methods from different biology views as baselines, including a sequence alignment-based GO prediction method (SAGP, see S1 Text), a PPI-based GO predictor (PPIGP, S2 Text), and a Naïve-based GO predictor (NGP, see S3 Text). In addition, we compared ATGO with five state-of-the-art models, including three deep learning-based methods (i.e., DeepGO [9], DeepGOCNN [21], and TALE [22]), two template-based methods (FunFams [31] and DIAMONDScore [21]), which are driven by protein family search and sequence homology alignment, respectively. Given the complementarity of machine learning and homology-based alignments, we also implemented a composite method, ATGO+, which generates predictions by linearly combining the confidence score of ATGO and SAGP. This will be benchmarked with two other composite methods (DeepGOPlus [21] and TALE+ [22]), which are weighted combinations of (DeepGOCNN, DIAMONDScore) and (TALE, DIAMONDScore), respectively. For the seven third-party methods, we downloaded the programs and ran them on our test dataset under the default setting.

Table 1 summarizes the overall results of all 12 GO prediction methods on 1068 non-redundant test proteins (See 'Methods and materials") for three GO aspects, where the performance is measured by maximum $F_1$-score ($F_{max}$) [32,33], area under the precision-recall curve (AUPR) [34], and coverage [33]. The data shows that ATGO achieves a significantly better performance than all other 8 non-composite control methods. Specifically, compared with the second-best performer SAGP, ATGO gains 9.3% and 69.1% average improvements for $F_{max}$ and AUPR, respectively, on the average of three GO aspects, all with *p*-values ≤1.1e-07 on two-sided Student's t-test.

After combining ATGO with SAGP, the composite AGTO+ achieves a small but consistent improvement over all three categories of GO aspects in terms of $F_{max}$ and AUPR, where the $p$-values in Student's t-test between them are both below 4.3e-02, showing the difference is statistically significant at the entire dataset level. ATGO+ also significantly outperforms the two composite control methods, DeepGOPlus and TALE+, with $p$-values below 8.4e-09 in all the comparisons, although these two control methods clearly outperform their corresponding single-based methods (DeepGOCNN and TALE) respectively. It cannot escape our notice that the magnitude of performance difference between ATGO and ATGO+ is smaller than that between TALE and TALE+ (or between DeepGOCNN and DeepGOPlus) in each GO aspect as shown in Table 1. Part of the reason is that ATGO by itself is a much more accurate predictor compared to other deep learning predictors such as DeepGO and TALE. Therefore, adding another component of sequence homology can bring a relatively smaller increase in the overall $F_{max}$ and AUPR scores. In fact, ATGO alone already outperforms both DeepGOPlus and TALE+, which provides a solid base for the better performance of AGTO+. Nevertheless, a consistent improvement has been seen in all datasets and approaches when adding the homology transferal component, demonstrating the advantage of combining different sources of information for improving protein function predictions.

It is noted that the Student's t-test $p$-values in Table 1 are calculated by comparing 10 independent implementations of ATGO/ATGO+ to a single implementation of the competing methods. To examine the statistical significance between all individual methods, most of which have only single implementation, we perform Friedman test [35], one of the most used approaches in analysis of variance, for the 12 GO prediction methods at individual protein level (see details in S4 Text). A significant performance difference with $p$-values $\leq$4.6e-106 is found among all prediction methods in three GO aspects. Thus, a Nemenyi post-hoc test [36] is performed to calculate the $p$-value of performance difference between each pair of prediction methods, with result listed in S1 Table. It can be found that the $p$-value between the propose ATGO/ATGO+ and each of competing methods is below 0.05 in each GO aspect at the individual protein level, except for (ATGO, SAGP), (ATGO, DeepGOPlus), and (AGTO+, SAGP) in MF with the $p$-values of 2.5e-01, 3.0e-01, and 1.5e-01, respectively. However, the $p$-values of $F_{max}$ values for the above-mentioned three pairwise methods in MF are 1.1e-07, 1.5e-06, and 4.8e-11, respectively, under Student's t-test at the entire dataset level, while the corresponding $p$-values for AUPR values are 3.1e-17, 1.1e-11, and 1.6e-18, respectively, as shown in Table 1. Meanwhile, although the difference between ATGO and ATGO+ is not significant at the individual protein level with the Nemenyi post-hoc test $p$-value$>$ = 0.90 for three GO aspects, the difference is statistically significant at the entire dataset level, with the Student's t-test $p$-value = 1.3e-03/1.2e-12/3.2e-05 for $F_{max}$ and 1.8e-09/3.9e-12/4.3e-02 for AUPR on MF/BP/CC terms respectively. Part of the reason for the $p$-value difference in the two calculations is that at the individual protein level ATGO/ATGO+ are measured by a set of $F_1$-scores on hundreds of proteins with significant functional differences while at the entire dataset level the ATGO/ATGO+ are run at a fixed index obtained through the average of confidences scores of 10 models, where each ATGO+ model consistently shows higher evaluation index than the corresponding ATGO model. Another reason is that the Student's t-test is mathematically different from that in the Nemenyi post-hoc test (see explanation in S2 Table and S5 Text). Given that the Student's t-test can be approximated in a more precise range and not be affected by the performances of other prediction methods in the same group (see details in S5 Text), we present the results mainly on the t-test in this work.

It is noted that the $F_{max}$ values of TALE listed in Table 1 are considerably lower than those reported by the TALE paper [22]. The major reason for this discrepancy is that in the TALE paper, $F_{max}$ calculation includes the root GO terms of three GO aspects for both ground truth

and prediction, while in this study, we followed the standard practice of CAFA assessment [33] and excluded the root terms. We further re-calculated the $F_{max}$ and AUPR values with including root GO terms for ATGO and TALE, as shown in S3 Table, where ATGO achieves significantly higher $F_{max}$ and AUPR than TALE for all GO aspects regardless of whether root terms are included.

Table 1 also shows that the four methods (i.e., SAGP, PPIGP, FunFams, and DIAMOND-Score) are associated with lower coverage scores; this is because these methods fail to search available templates or interaction partners for some of the test proteins and therefore cannot provide the predictions for them. Since the lack of prediction may have impact on their overall prediction performances on the whole test dataset, we further benchmarked the 12 methods on a subset of 562 test proteins, for which predictions can be generated by all methods. As shown in S4 Table, a similar trend is observed in which ATGO/ATGO+ outperform the control methods with a significant marge. Meanwhile, it is observed that SAGP and DIAMOND-Score share an obviously higher prediction accuracy than PPIGP and FunFams for both tests, showing that sequence homology is more effective than PPI and family similarity for protein functional references.

We further assess the modeling results using an information theory-based evaluation metric, i.e., information content-weighted maximum $F_1$-score (ICW-$F_{max}$), which is defined by Eq S7 in S6 Text. The results for 12 GO prediction methods on the 1068 test proteins are listed in S5 Table, which shows again that the proposed ATGO and ATGO+ outperform the 10 competing methods using this new metric score.

## ATGO shows great generality to new species and rare GO terms

Despite the power of modeling, many of the machine-learning based methods can have reduced performance on the proteins from new species not included in their training dataset. To examine the generalizability of ATGO to new species, we mapped every protein in our dataset to the corresponding species and collected 160 test proteins from 104 new species which are never seen in training and validation datasets. In Fig 2, we listed the performance of ATGO/ATGO+ in control with other 9 GO prediction methods on the 160 proteins. Here, PPIGP was excluded because it cannot make any prediction for the above-mentioned 160 proteins as they have no available PPI information in STRING database [37].

From the view of $F_{max}$, ATGO and ATGO+ are top-two performers in MF and CC. As for BP, ATGO outperforms all single-based methods but slightly underperforms two composite methods (i.e., DeepGOPlus and TALE+), but ATGO+ achieves a better performance with $F_{max}$ 4.3% and 3.8% higher than the two composite methods. In terms of AUPR values, ATGO/ATGO+ is ranked 3/1, 2/1, and 1/2 for MF, BP, and CC, respectively, among all 11 methods. Meanwhile, the $F_{max}$/AUPR values of ATGO/ATGO+ for the 160 proteins in Fig 2 are largely comparable to that for the whole dataset in Table 1. These observations show that the performance of ATGO and ATGO+ does not degrade when modeling new species proteins, demonstrating the generalizability of the approaches.

Due to the nature of knowledge-based training, another challenge to the machine learning approaches is the modeling of rarely seen function terms. To examine ability of ATGO/ATGO+ on the GO terms of different popularities, we grouped GO terms in the test dataset into four groups: 5–10, 10–30, 30–50 and >50, in terms of the number of annotated proteins per GO term; these ranges correspond to the numbers of terms of 425, 328, 64 and 105, respectively. In Fig 3, we present the AUROC scores for the rare GO terms in ranges 5–10 and 10–30, respectively, where the mean and median AUROC scores are listed in S1 and S2 Figs. It is found that ATGO/ATGO+ are ranked as the top two methods. Taking range 5–10 as an example,

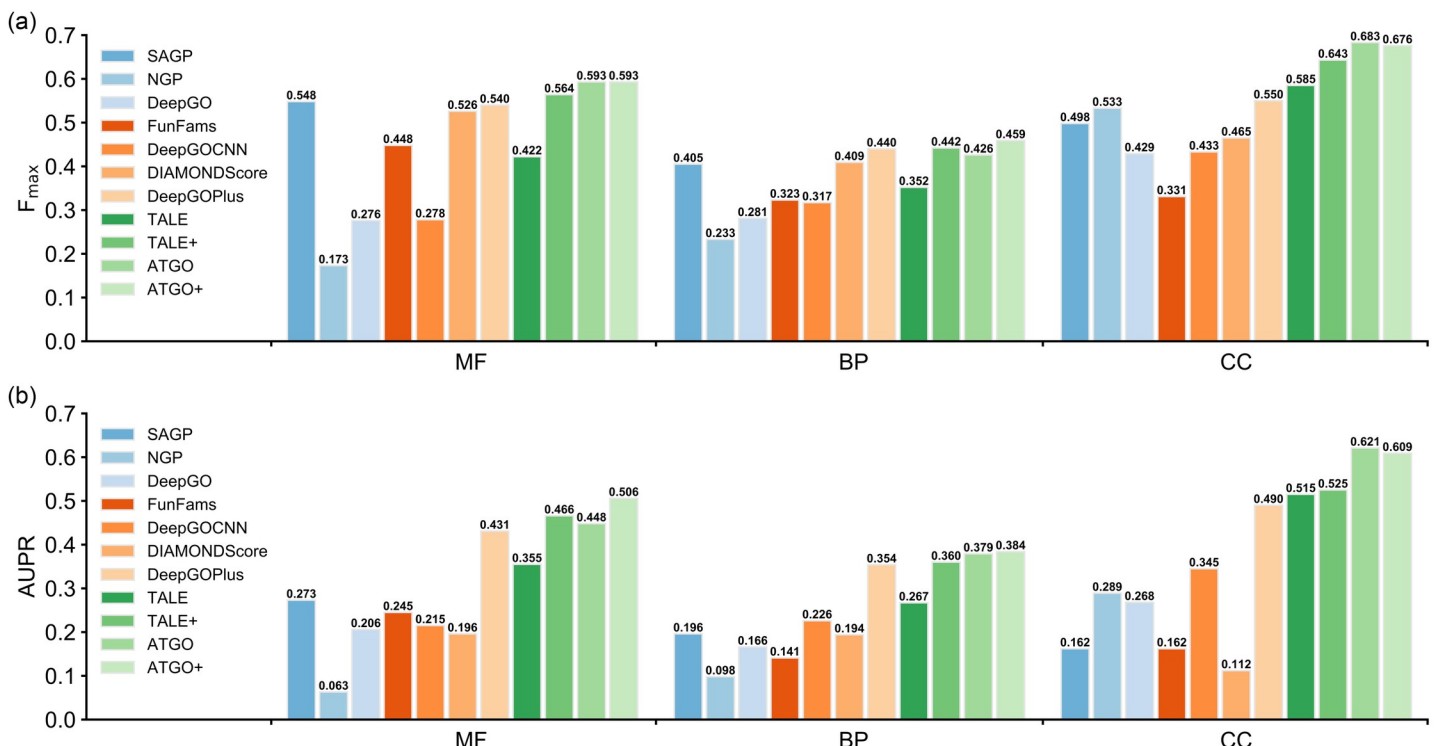

**Fig 2.** Comparison of ATGO/ATGO+ with 9 competing methods on the 160 test proteins from 104 new species for **(a)** $F_{max}$ and **(b)** AUPR.

ATGO+ shows better performance than all other 10 methods for each GO aspect. Specifically, ATGO+ gains 2.5% and 3.9% average improvements on three aspects for mean and median AUROC values, respectively, compared to the best of the control method (TALE+). For the range 10–30, ATGO and ATGO+ share the highest mean and median AUROC values both in BP and CC. As for MF aspect, ATGO+ and ATGO are ranked as 1/3 and 1/2 in terms of mean and median AUROC values, respectively.

In S3 Fig, we also list the results of the two more common GO terms with ranges of 30–50 and >50. ATGO and AGTO+ again outperform other control methods. These results demonstrate a balanced performance for all type of functional terms.

## Testing on targets of the third CAFA challenge (CAFA3)

As an independent test, we applied ATGO/ATGO+ to the third Critical Assessment of Protein Function Annotation (CAFA3) dataset which consists of 3328 test proteins [33]. To make a fair comparison, we re-trained the ATGO pipeline on a subset of the 66,841 training proteins that were released by the CAFA3 organizers. To remove homology contamination, we have filtered out the homologous proteins from the training dataset which have more than $t_1$ sequence identity to the test proteins. Here, we have randomly selected 95% of the filtered training proteins to re-train the ATGO model and used the remaining 5% proteins as the validation set to optimize the ATGO parameters. For the in-house SAGP, PPIGP, and NGP programs, these non-homology training proteins were used to construct the template databases and prior probabilities of GO terms.

Table 2 summarizes the performance of the 10 GO prediction methods on all of 3328 CAFA3 test proteins under the cut-off $t_1 = 30\%$. Here, we excluded TALE and TALE+ from

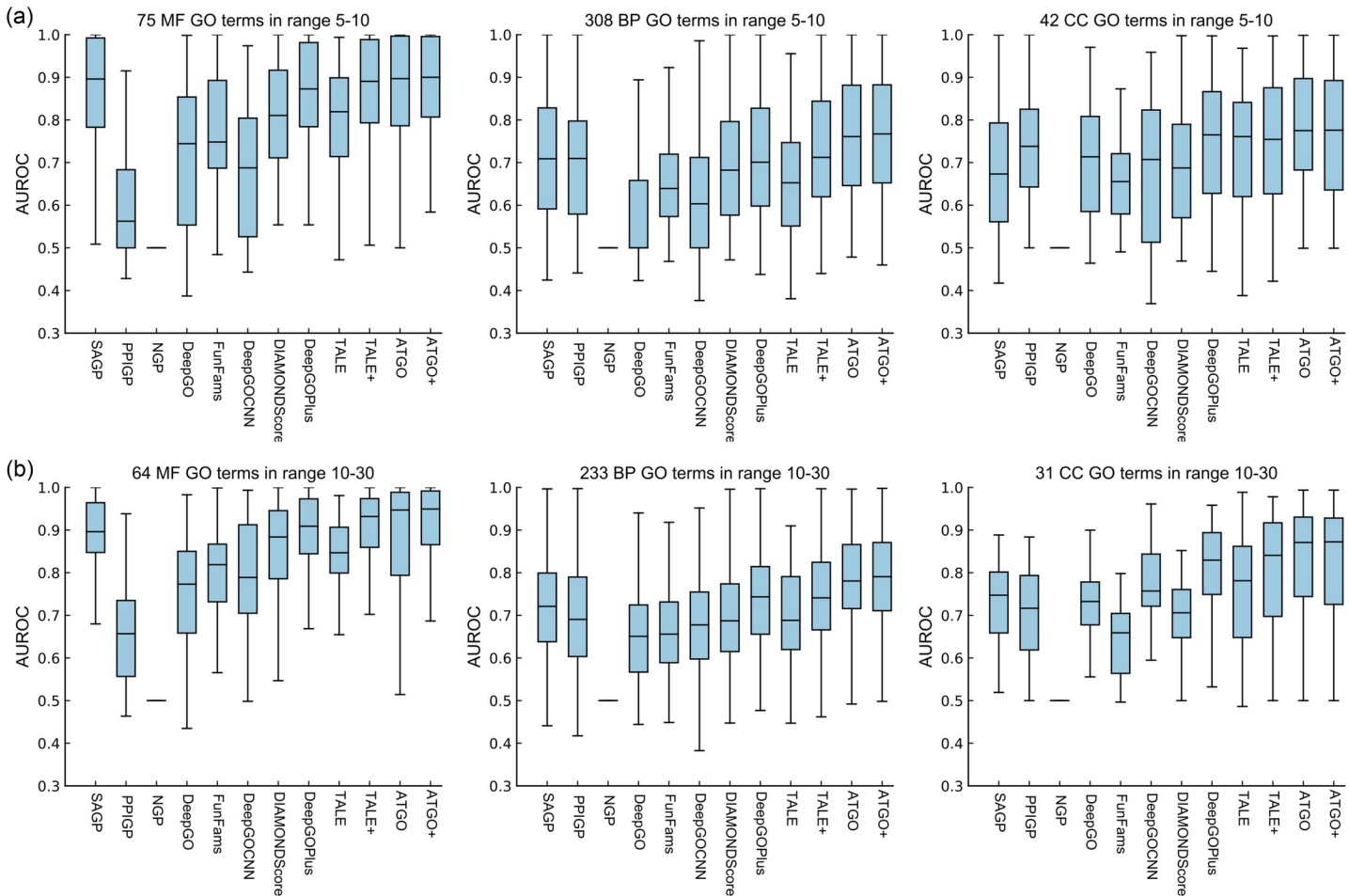

**Fig 3. AUROC values by different methods for the rare GO terms in terms of the number of associated proteins, where the median line in the box is the median AUROC value. (a)** range 5–10. **(b)** range 10–30.

the method list because the models trained on the CAFA3 dataset are unavailable for TALE. It is noted that for other third-party methods, the homologous proteins were not removed from the training dataset and run with the default setting. The result shows that ATGO and ATGO + outperform other single and composite methods, respectively, for every GO aspect. Taking ATGO as an example, it achieves 8.2%, 6.5%, and 14.6% improvements of $F_{max}$ values for MF, BP, and CC, respectively, compared to the best single method (SAGP), all with $p$-values below 2.2e-07 under Student's t-test.

To examine the pairwise significance, we performed Friedman test [35] for the 10 GO prediction methods at the individual protein level and found that there exists a significant performance difference with $p$-value $\leq$6.0e-168 in each GO aspect. Next, the Nemenyi post-hoc test [36] is further performed to calculate the $p$-values of performance difference between all method pairs. As shown in S6 Table, the $p$-values between ATGO/ATGO+ and all the competing methods are both below 0.05 in three aspects at the individual protein level. In addition, we list in S7 Table the ICW-$F_{max}$ scores performed by all 10 GO prediction methods, where ATGO/ATGO+ show again a better performance than other competing methods in all three GO aspects.

**Table 2. Comparison of ATGO/ATGO+ with other 8 competing methods on 3328 CAFA3 targets where a sequence identity cut-off $t_1$ = 30% between the training and testing proteins was applied to the five in-house methods (ATGO, ATGO+, SAGP, PPIGP, and NGP).** *p*-values in parenthesis are calculated between ATGO and other single-based methods and between ATGO+ and other composite methods by two-sided Student's t-test. Specifically, the proposed ATGO and ATGO+ are repeatedly implemented with 10 times on the benchmark dataset to generate the corresponding performance evaluation indices, which are compared with the fixed evaluation index generated by the competing method to calculate *p*-value using two-sided Student's t-test. Bold fonts highlight the best performer in each category. Coverage is the ratio of the number of proteins with available prediction scores divided by the total number of test proteins.

| Methods | | $F_{max}$ | | | AUPR | | | Coverage | | |
|---|---|---|---|---|---|---|---|---|---|---|
| | | MF | BP | CC | MF | BP | CC | MF | BP | CC |
| Single algorithms | SAGP | 0.463 | 0.465 | 0.473 | 0.244 | 0.302 | 0.298 | 0.82 | 0.90 | 0.85 |
| | | (4.3e-10) | (2.2e-07) | (3.2e-11) | (9.8e-19) | (2.2e-14) | (6.4e-19) | | | |
| | PPIGP | 0.248 | 0.377 | 0.453 | 0.153 | 0.296 | 0.421 | 0.89 | 0.88 | 0.84 |
| | | (5.6e-18) | (3.2e-13) | (3.0e-12) | (4.5e-20) | (1.2e-14) | (3.9e-16) | | | |
| | NGP | 0.159 | 0.302 | 0.445 | 0.066 | 0.170 | 0.366 | 1.00 | 1.00 | 1.00 |
| | | (3.6e-19) | (3.4e-15) | (1.4e-12) | (4.9e-21) | (7.3e-18) | (1.3e-17) | | | |
| | DeepGO | 0.275 | 0.386 | 0.487 | 0.198 | 0.291 | 0.487 | 1.00 | 1.00 | 1.00 |
| | | (1.6e-17) | (6.8e-13) | (2.8e-10) | (1.8e-19) | (7.8e-15) | (6.3e-13) | | | |
| | FunFams | 0.470 | 0.428 | 0.464 | 0.304 | 0.228 | 0.284 | 0.65 | 0.71 | 0.66 |
| | | (4.7e-09) | (6.4e-11) | (1.0e-11) | (1.6e-17) | (1.1e-16) | (3.8e-19) | | | |
| | DeepGOCNN | 0.311 | 0.291 | 0.413 | 0.231 | 0.191 | 0.288 | 1.00 | 1.00 | 1.00 |
| | | (8.0e-17) | (2.0e-15) | (9.8e-14) | (5.9e-19) | (1.8e-17) | (4.4e-19) | | | |
| | DIAMONDScore | 0.456 | 0.450 | 0.464 | 0.199 | 0.268 | 0.238 | 0.76 | 0.85 | 0.80 |
| | | (8.8e-11) | (3.2e-09) | (1.1e-11) | (1.9e-19) | (1.3e-15) | (8.6e-20) | | | |
| | ATGO | **0.501** | **0.495** | **0.542** | **0.469** | **0.397** | **0.546** | 1.00 | 1.00 | 1.00 |
| Composite algorithms | DeepGOPlus | 0.459 | 0.460 | 0.474 | 0.392 | 0.342 | 0.470 | 1.00 | 1.00 | 1.00 |
| | | (9.2e-12) | (4.5e-13) | (4.0e-12) | (2.3e-15) | (3.4e-15) | (3.8e-14) | | | |
| | ATGO+ | **0.511** | **0.502** | **0.543** | **0.477** | **0.412** | **0.546** | 1.00 | 1.00 | 1.00 |

Among the 3328 test proteins, 1177 have no knowledge on any of the GO aspects before the CAFA3 experiment, where other 2151 proteins have limited knowledge on some GO terms and with other GO terms determined during the CAFA3 experiment. In S8 Table, we list the GO prediction results on the no-knowledge (NK) and limited-knowledge (LK) datasets separately. The ATGO/ATGO+ outperform again other existing methods in all GO aspects for both datasets, showing that the superiority of the pipeline does not depend on specific protein datasets.

Finally, we examine our methods for different species in CAFA3 test dataset. According to statistics, the 3338 test proteins are originated from 20 species, as shown in S9 Table, where Human, Arabidopsis, Fission Yeast, and Mouse are the top-four species in terms of the sample number. Fig 4 lists the $F_{max}$ values of the 10 GO prediction methods for the four species under the cut-off $t_1$ = 30%, while the corresponding AUPR values are given in S4 Fig. For Human, Arabidopsis, and Mouse, ATGO and ATGO+ achieve the highest $F_{max}$ values among 10 methods for all three GO aspects. Taking Human species as an example, the $F_{max}$ values of ATGO + are 2.1%, 2.1%, and 12.2% higher than that of the third-best performer in MF, BP, and CC, respectively. In Arabidopsis and Mouse, the improvements between ATGO/ATGO+ and the control methods are more significant. As for Fission Yeast, although ATGO/ATGO+ show a slightly lower $F_{max}$ in comparison with DeepGOPlus and DIAMONDScore for MF, they achieve the highest $F_{max}$ for both BP and CC aspects.

In addition, the prediction performance of the ATGO/ATGO+ under the cut-off $t_1$ = 100% on CAFA3 test dataset is summarized in S10 Table, where $t_1$ = 100% indicates that we did not filter out any homologs from the training dataset. Since the third-party programs did not remove homologs, we only listed the in-house programs. Again, the ATGO/ATGO+ show

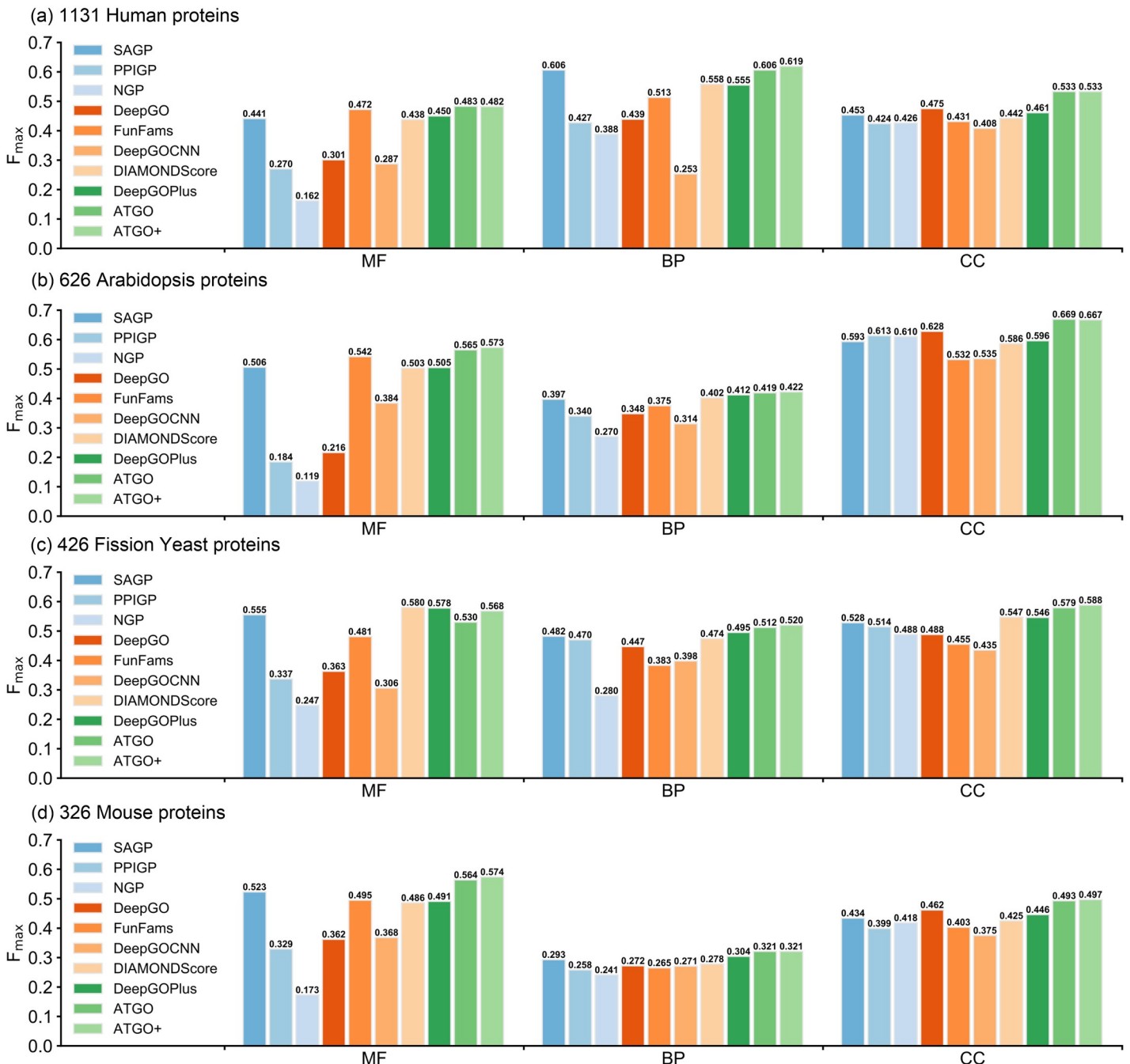

**Fig 4. The $F_{max}$ values of 10 GO prediction methods under the homology cut-off $t_1$ = 30% between training and testing proteins for four individual species in CAFA3 test dataset. (a)** Human; **(b)** Arabidopsis; **(c)** Fission Yeast; **(d)** Mouse.

superior performance in comparison with the competing methods for each GO aspect. It is also observed that ATGO has a lower sensitivity on homologous training proteins than the sequence-based method of SAGP. Specifically, after reducing cut-off $t_1$ from 100% to 30%, the $F_{max}$ and AUPR values of ATGO are separately decreased by 5.2% and 6.2% on average of three GO aspects, while the corresponding decreases for SAGP are 8.9% and 19.3%. This is not

surprising because the sequence alignment-based approach can obtain closer homology templates and therefore higher accuracy predictions when given a higher sequence identity cut-off.

Interestingly, SAGP outperforms most of other control methods. Although both are built on BLAST alignments, SAGP is different from the BLAST baseline method used in CAFA challenge [33]; the main difference between these two is that the former deduces consensus functional patterns from multiple homology templates (see Eq S1 in S1 Text) while the latter only uses a single template for function deduction (see Eq S10 in S7 Text). In S11 Table, we further compare SAGP with the CAFA BLAST baseline in our constructed test dataset (1068 non-redundant proteins) and CAFA3 test dataset, respectively, with details explained in S7 Text. It is shown that SAGP significantly outperforms the CAFA BLAST baseline, suggesting the importance of combining multiple templates in the homology-based protein function inference. Consistently with the CAFA experiments [33], the BLAST baseline also underperforms other competing methods, such as FunFams and DeepGOPlus, as shown in Tables 1, 2 and S11.

## Ablation study

To analyze the contributions of algorithmic innovations in ATGO to its improved performance, we design an ablation study, in which we start from a baseline model (M0) and incrementally add algorithmic components of ATGO to build three advanced models (M1, M2 and M3, with M3 = ATGO). The pipelines of the four models are designed as follows (see S5 Fig for the architectures):

- **M0**: Model is trained with the standard CNNs with a one-hot coding matrix extracted from the input sequence, followed by a fully connected network, in which an output layer with sigmoid activation function [38] is added to generate the confidence scores of predicted GO terms. In the training stage, the cross-entropy loss [39] is used as the loss function, as described in Eq 5.

- **M1:** We replace the CNN by the ESM-1b transformer and extract the feature embeddings from the last layer for input sequence, which is further fed to a fully connected network with sigmoid activation function to output the confidence scores of predicted GO terms, where the cross-entropy loss is used as loss function in the training stage.

- **M2:** We add the triplet network-based guilt-by-association (GBA) strategy in M1, where the loss function is the combination of triplet loss [30] and cross-entropy loss, as shown in Eq 4. The final outputs are the combination of confidence scores from triplet network-based GBA strategy and those from the output layer with sigmoid activation function, as described in Eq 1.

- **M3:** We add the multi-view feature fusion strategy in M2 to build the final model, where the embedding features are derived from the last three layers of ESM-1b rather than only the last layer (see "Methods and materials").

Fig 5 illustrates the $F_{max}$ and AUPR values of four models for three GO aspects on all of 1068 test proteins, where we run each model for 10 times and then used the average of all predictions as the final result. Compared with M0, M1 achieves a significant gain with the average $F_{max}$ and AUPR increased by 32.4% and 41.8%, respectively, demonstrating that ESM-1b transformer is critical to improve function prediction of the ATGO pipeline. Compared to other function categories, M1 achieves the highest performance improvement in MF prediction, indicating that the embeddings from ESM-1b contain sequence signals which have a closer relationship with MF than with BP and CC. This is probably because MF is mainly

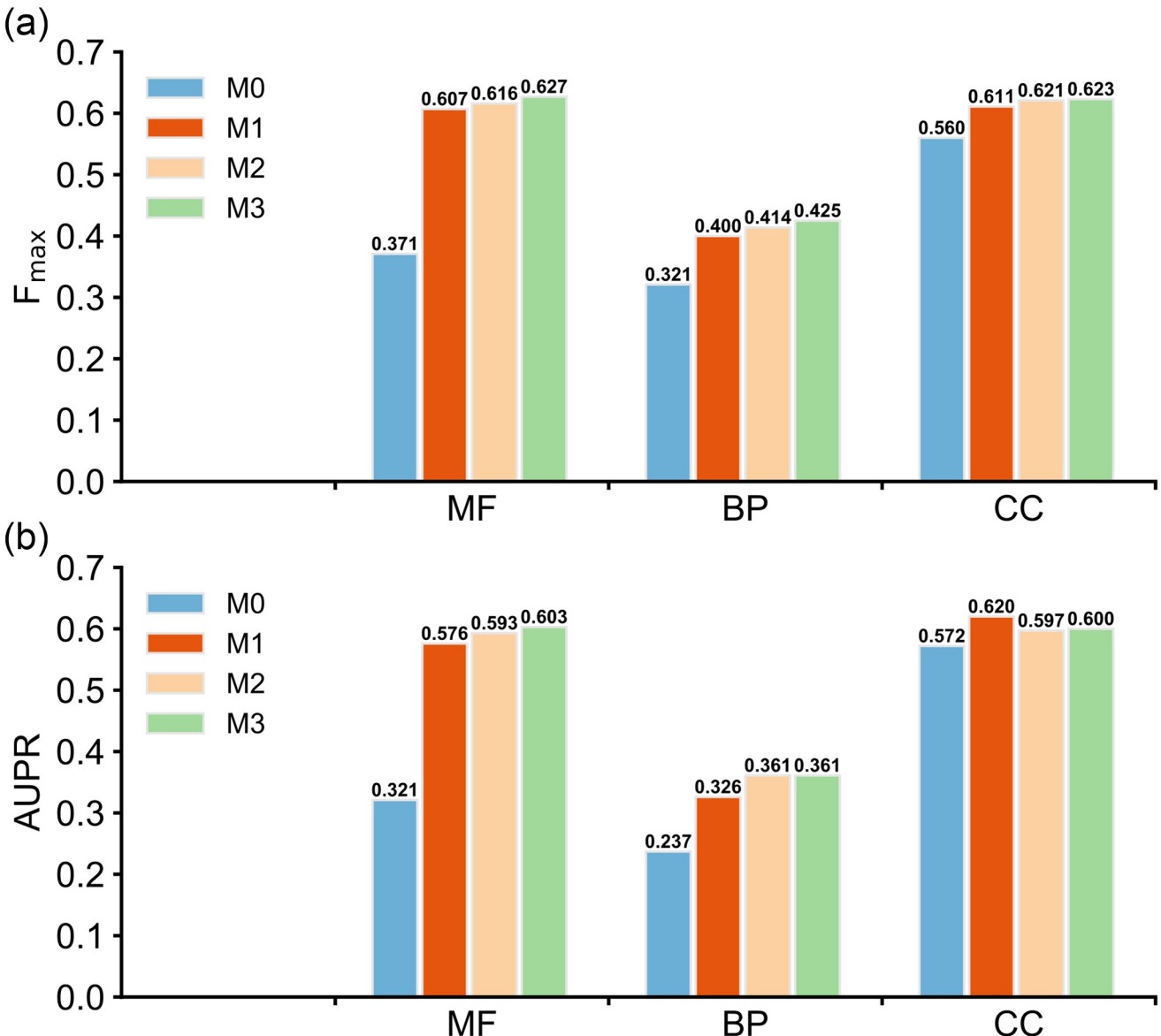

**Fig 5. Ablation study on the ATGO pipeline based on 1068 test proteins.** M0 denotes the baseline model with one-hot embedding, where M1-3 are the models with transformer embedding, triplet network, and multi-view feature added respectively. **(a)** The $F_{max}$ values of four models. **(b)** The AUPR values of four models.

associated with the molecular action that a protein can perform by itself and therefore directly related to the internal feature of the sequence that can be captured by the sequence embedding. On the other hand, BP describes the biological pathways which often involve a series of inter-molecular actions. Rather than a single protein, a group of interacting proteins are required to complete a BP. Since the sequence embedding is primarily derived from the sequence itself and not designed for capturing the information of protein-protein interactions or co-expression regulations, it is less effective for BP prediction. Similarly, the CC of a protein such as sub-cellular localization is dependent on other molecules in the cell and cannot be fully captured by the embedding.

After separately adding triplet network-based GBA strategy to M1 and multi-view feature fusion strategy to M2, the corresponding $F_{max}$ values are increased on average by 2.2% and 1.6% on three GO aspects. The AUPR values of M2 and M3 are slightly decreased for CC aspect in comparison with M1, but the corresponding values are sustainably increased in other two aspects with $p$-values$\leq$9.1e-18 after adding the two strategies. Although M2 and M3 use the same ESM-1b model for feature extraction and the same loss function for training, M3 still slightly outperforms M2 by using the last three rather than just the last one layer of ESM-1b for feature extraction. This is partly because that different layers of a transformer model such as ESM-1b capture different levels of information and the layers closer to the end of the transformer tend to extract more abstract information. Therefore, M3 captures more fine-grained information from ESM-1b than M2. These observations indicate that the additional two strategies are helpful for enhancing the overall performance of function prediction, although less significant than the ESM-1b transformer.

In addition, to examine the impact of different metric learning methods on the ATGO model (see "Methods and materials"), we used four metrics of $F_1$-score, Jaccard similarity [40], weighted $F_1$-score, and weighted Jaccard similarity to assess the functional similarity in the triplet loss separately, where the weights of GO terms are measured by information content [41]. The performance of the ATGO models via the four metric learning methods on our test datasets is summarized in S12 Table and discussed in S8 Text. It is found that although the performance of individual models varies, there is no obvious difference on the overall performance of the models for each GO aspect, suggesting that the effectiveness of the proposed ATGO framework is not sensitive to the choices of different metric learning methods.

## Case studies

To further examine the effects of different GO prediction methods, we selected three representative proteins from our test dataset (**A6XMY0**, **E7CIP7**, and **F4I082**) as illustrations. These proteins are associated with 18, 14, and 13 GO terms, respectively, for the BP aspect in the experimental annotation, where the root GO term (GO: GO:0008150, biological process) is excluded. Table 3 summarizes the numbers of correctly predicted GO terms (i.e., true positives) and mistakenly predicted terms (i.e., false positives), and the $F_1$-scores between predicted and native GO terms (see Eq S32 in S10 Text) in the BP prediction for three proteins by 12 GO prediction methods.

**Table 3. The modeling results of the ATGO/ATGO+ in control with other 10 competing GO prediction methods on three representative examples in biological process (BP) prediction, where bold fonts highlight the best performer in each category.**

| Methods | | A6XMY0 | | | E7CIP7 | | | F4I082 | | |
|---|---|---|---|---|---|---|---|---|---|---|
| | | TP | FP | $F_1$-score | TP | FP | $F_1$-score | TP | FP | $F_1$-score |
| Single algorithms | SAGP | 0 | 0 | 0.000 | 8 | 9 | 0.516 | **13** | 8 | 0.765 |
| | PPIGP | 0 | 0 | 0.000 | 0 | **0** | 0.000 | 3 | 16 | 0.187 |
| | NGP | 2 | 16 | 0.111 | 4 | 14 | 0.250 | 1 | 17 | 0.065 |
| | DeepGO | 14 | 29 | 0.459 | 0 | 20 | 0.000 | 2 | 57 | 0.056 |
| | FunFams | 0 | 0 | 0.000 | 0 | 14 | 0.000 | 0 | **0** | 0.000 |
| | DeepGOCNN | 14 | 26 | 0.483 | 3 | 9 | 0.231 | 1 | 11 | 0.080 |
| | DIAMONDScore | 0 | 0 | 0.000 | 8 | 9 | 0.516 | 0 | **0** | 0.000 |
| | TALE | 14 | 5 | 0.757 | 10 | 2 | 0.769 | 1 | 9 | 0.087 |
| | ATGO | **15** | 0 | **0.909** | **14** | 2 | **0.933** | 9 | 7 | **0.621** |
| Composite algorithms | DeepGOPlus | 1 | 2 | 0.095 | 8 | 10 | 0.500 | 0 | 1 | 0.000 |
| | TALE+ | 1 | 0 | 0.105 | 8 | 9 | 0.516 | 0 | 1 | 0.000 |
| | ATGO+ | **11** | 0 | **0.759** | **14** | 2 | **0.933** | **13** | 3 | **0.897** |

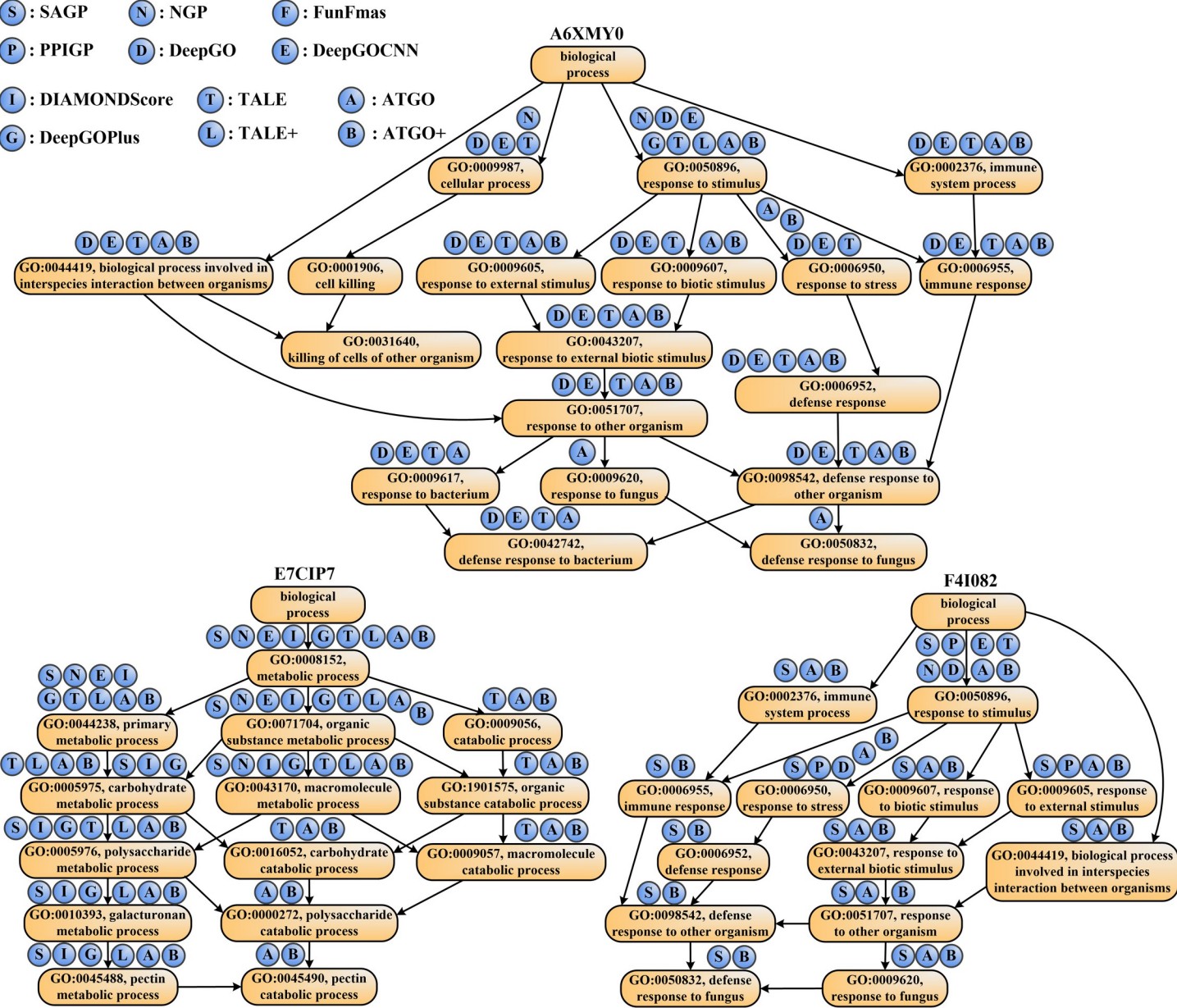

**Fig 6. The directed acyclic graph with GO terms for three representative examples of A6XMY0, E7CIP7, and F4I082 in biological process.** The circles above each GO term refer to the prediction methods, where a circle filled with "X" on GO term "Y" indicates that method "X" can correctly predict term "Y".

In Fig 6, we plot the directed acyclic graph of GO terms in the experimental annotation and the correctly predicted GO terms of 12 methods for three proteins. Moreover, the false positives of each method are listed for all three examples in S13 Table. It should be noted that the predicted GO terms of different methods are determined by their own cut-off setting to achieve the highest $F_{max}$ value.

Several interesting observations can be made from the data. First, for two of three proteins (i.e., **A6XMY0** and **E7CIP7**), the proposed ATGO and ATGO+ are top-two performers with the highest $F_1$-scores. In **A6XMY0**, for example, the four template-based or biology network-based methods, including SAGP, PPIGP, FunFams, and DIAMONDScore, cannot make any

predictions, because they fail to hit the available sequence templates or PPIs. At the same time, ATGO correctly recognizes the most (15) GO terms with no false positives, indicating that the deep learning-based ATGO has low dependency on the availability of templates and interaction partners.

For **E7CIP7**, all true positive predictions by the 10 control methods can be effectively identified by ATGO. More importantly, ATGO can correctly recognize two additional GO terms, i.e., GO: 0000272 and GO:0045490, which are missed by all the 10 control methods. This observation shows that ATGO can predict functions in a more precise level, due to the fact that it successfully identifies several children GO terms, in which other methods fail. However, because SAGP cannot provide additional true positives for ATGO, the performance of composite ATGO+ cannot be further improved and even be slightly degraded in **A6XMY0**. This example shows that although homology-based transferals can help improve the overall performance of ATGO+, it could negatively impact the modeling accuracy when the template quality is poor. It might be helpful to introduce additional filters to the component of homology-based models, e.g., on the confidence score of sequence alignments.

Occasionally, ATGO shows the worse performance with a lower $F_1$-score in comparison with SAGP, as illustrated in **F4I082**. For this case, SAGP generates the highest number of true positives among all 9 single-based methods, where four GO terms (i.e., GO:0006955, GO:0006952, GO:0098542, and GO:0050832) are missed by ATGO. Due to the valuable information inherited from SAGP, the composite ATGO+ obtains the highest $F_1$-score across all single and composite methods, which demonstrates the advantage of ATGO+ by combining composite homology and deep learning-based results.

## Discussion

We developed a new deep learning-based method, named ATGO, to predict functions of proteins from the primary sequence. The algorithm was built on transformer embedding and triplet network decoding. The large-scale tests on a set of 1068 non-redundant benchmark proteins and 3328 targets from the community-wide CAFA3 experiments demonstrated that ATGO consistently outperforms other state-of-the-art approaches in the accuracy of gene-ontology (GO) predictions. The improvement of ATGO can be attributed to several advancements. First and most importantly, the ESM-1b transformer can effectively extract the discriminative feature embeddings for the input sequence from the different views of evolution diversity. Second, the multi-view feature fusion helps reduce the negative impact caused by information loss in feature extraction. Third, the triplet network-based GBA strategy is important to enhance the training efficiency by maximizing the feature distance between positive and negative samples. Finally, combining ATGO with complementary information from sequence homologous inference can further improve the prediction accuracy.

Despite the demonstrated effectiveness of the transformer, it is important to note that the currently used ESM-1b transformer is only one of the several existing language models built on a single query sequence. The use of other embedding language models such as ProtTrans and a newly released extended version ESM-2 [42], both of which demonstrated superiority to ESM-1b, for GO prediction is worthy of exploration in future work. Moreover, given that multiple sequence alignment (MSA) contains significantly more enriched evolutionary information labelled with conserved protein sites [43], which are critical to protein functional annotations, we will construct a new unsupervised protein language model by the use of the MSAs created from DeepMSA [44] through self-attention networks. The embedding from MSA transformers should help further improve the sensitivity and accuracy of the GO prediction models [45]. Studies along these lines are under progress.

## Methods and materials

### Dataset construction

To construct the datasets, we downloaded all protein entries from the Gene Ontology Annotation (GOA) database [46], which provides functional annotations for the sequences in the UniProt database [4]. To ensure the reliability, we only considered the 123,774 proteins whose functions are annotated by at least one of the eight experimental evidence codes (i.e., EXP, IDA, IPI, IMP, IGI, IEP, TAS, and IC) [47,48].

Among the 123,774 proteins, we selected 1068 non-redundant proteins as the test dataset, which have the function annotations deposited in the manually reviewed Swiss-Prot library [49] after 2019-1-1; and 1089 non-redundant proteins as the validation dataset, which were deposited in Swiss-Prot from 2017-1-1 to 2018-12-31. The remaining 109,132 proteins were used as the training dataset of ATGO. Here, a sequence identity cut-off 30% has been used to filter out the redundant proteins within each dataset and between different datasets using the CD-HIT [50] program. The number of entries in each dataset on different GO categories is summarized in S14 Table. While the training and validation datasets were separately used to train the ATGO models and optimize the parameters, the test dataset was used to evaluate the performance of the models.

### The framework of ATGO

ATGO is a deep learning-based protein function prediction method, with input being a query amino acid sequence and output including confidence scores of predicted GO terms. As showed in Fig 1A, ATGO consists of three procedures of multiple-view feature extraction using transformer, neural network-based feature fusion, and triplet network-based function prediction, where the first and second procedures are jointly defined as feature generation model (FGM).

**Procedure I: Multiple-view feature extraction using transformer.** The input sequence is fed to ESM-1b transformer with 33 attention layers to extract the feature embeddings. Specifically, each layer outputs a feature embedding from an individual evolutionary view, which is further used as the input of next layer to generate a new embedding from a more complex view. Considering that the feature embedding from a single view (layer) cannot fully represent the evolutionary information for sequence, we extract feature embeddings from multiple views (i.e., the last three layers) to relieve information loss. Each embedding is represented as a $L \times D$ matrix, where $L$ is the length of query, and $D = 1280$ is a preset hyper-parameter in ESM-1b.

**Procedure II: Neural network-based feature fusion.** We calculate the average value for each column in the embedding matrix to generate an embedding vector with $D$ dimension, as the input of fully connected layer with $N_1$ neurons. Then, the outputs of three fully connected layers are concatenated as a new vector with $3N_1$ dimension, which is further fed to another fully connected layer (i.e., $FCL_a$) with $N_2$ neurons. Here, we set $N_1 = N_2 = 1024$.

**Procedure III: Triplet network-based function prediction.** The triplet network-based GBA strategy is performed on the output of $FCL_a$ to generate a confidence score vector $s_{gba}$ for predicted GO terms. At the same time, the output layer $FCL_O$ with sigmoid activation function [38] is fully connected to $FCL_a$ to output another confidence score vector $s_{saf}$. The final confidence score vector is the weight combination:

$$s = w s_{gba} + (1 - w) s_{saf} \tag{1}$$

where $w$ is the weight and ranges from 0 to 1.

## ESM-1b transformer

The architecture of ESM-1b transformer [26,51] is illustrated in Fig 1B. For an input sequence, the masking strategy [52] is performed on the corresponding tokens (i.e., amino acids). Specifically, we randomly sample 15% tokens, each of which is changed as a special "masking" token with 80% probability, a randomly-chosen alternate amino acid with 10% probability, and the original input token (i.e., no change) with 10% probability. Then, the masked sequence is represented as a $L \times 28$ matrix using one-hot encoding [53], where 28 is the types of tokens, including 20 common amino acids, 6 non-common amino acids (B, J, O, U, X and Z), 1 gap token, and 1 "masking" token.

The one-hot matrix is firstly embedded with positions and then fed to a self-attention block [54] with $n$ layers, each of which consists of $m$ attention heads, a linear unit, and a feed-forward network (FFN). In each head, the scale dot-product attention is performed on three matrices, including $M_Q$ (Query), $M_K$ (Key), and $M_V$ (Value), as follows. First, the dot-product between $M_Q$ and $M_K$ is performed to generate an $L \times L$ weight matrix, which measures the similarity for each amino acid pair in sequence. Then, we use the scale parameter and SoftMax function to normalize the weight matrix. Finally, the attention matrix is generated by multiplying the normalized weight matrix with $M_V$.

In each layer, all attention matrices are concatenated as a new matrix, which is further fed to the subsequent linear unit and FFN with shortcut connections to output feature embedding. The output of the last attention layer is fed to a fully connect layer with SoftMax function to generate a $L \times 28$ probability matrix $P$, where $P_{lc}$ indicates the probability that the $l$-th token in the masked sequence is predicted as the $c$-th type of amino acid.

The loss function is designed as a negative log likelihood function between inputted one-hot and outputted probability matrices, to ensure that the prediction model correctly predicts the true amino acids in the masked position as much as possible. The mathematics formulas of the above-mentioned procedures are listed in S9 Text.

The ESM-1b transformer is optimized by minimizing the loss function via Adam optimization algorithm [55]. Then, the output of each attention layer is a $L \times D$ feature embedding, where $D$ is the number of neurons of FFN. The current ESM-1b model was trained on 27.1 million proteins from UniRef50 database and can be download at https://github.com/facebookresearch/esm, where $n = 33$, $m = 20$, and $D = 1280$.

## Triplet network-based guilt-by-association for GO prediction

In GBA strategy, we select the templates with the most similar feature embeddings for a query from the training dataset to annotate the query, where the similarity of feature embeddings is measured by a supervised triplet network [30], as shown in Fig 1C.

The input is a triplet variable (*anc*, *pos*, *neg*), where *anc* is an anchor (baseline) protein, and *pos* (or *neg*) is a positive (or negative) protein with the same (or different) function of *anc*. Each sequence is fed into feature generation model (FGM, see Fig 1A) to generate a feature representation vector, as the input of $FCL_a$, to output a new embedding vector. Then, the feature dissimilarity between two proteins is measured by Euclidean distance [56] of embedding vectors. Finally, a triplet loss is designed to associate feature similarity with functional similarity:

$$Tripletloss = max(d(anc, pos) + margin - d(anc, neg), 0) \qquad (2)$$

where $d(anc, pos)$ (or $d(anc, neg)$) is the Euclidean distance of feature embeddings between anchor and positive (or negative), and *margin* is a pre-set positive value. Here, the minimization of triplet loss requests for the maximization of $d(anc, neg) - d(anc, pos)$. In the ideal case,

*Tripet loss* = 0 when $d(anc, neg) \geq d(anc, pos) + margin$, which indicates substantially higher feature similarity of anchor proteins to positives than to negatives.

We use the "batch on hard" strategy [57,58] to calculate the triplet loss for a training dataset:

$$Loss_t = E_{x \sim X} max(d(x, pos)_{max} + margin - d(x, neg)_{min}, 0) \tag{3}$$

where $x$ is a sequence in training set $X$, and $d(x, pos)_{max}$ (or $d(x, neg)_{min}$) is the maximum (or minimum) value of Euclidean distances between $x$ and all positive (or negative) proteins with same (or different) function of $x$. Moreover, two proteins are considered to have the same function if their functional similarity is larger than a cut-off value $c_f$. The functional similarity of two proteins is measured by $F_1$-score between their GO terms, as shown in S10 Text.

It has been demonstrated that the cross-entropy loss [39] helps to improve the performance of triplet network [59–61]. Therefore, we further designed a composite loss function for ATGO through combining triplet loss with cross-entropy loss:

$$Loss_{ATGO} = Loss_t + \alpha \cdot Loss_c \tag{4}$$

$$Loss_c = -E_{x \sim X} E_{q \sim Q} y(x, q) \cdot log s_{saf}(x, q) + (1 - y(x, q)) log(1 - s_{saf}(x, q)) \tag{5}$$

where $\alpha$ is a balanced parameter, $q$ is an element in candidate GO term set $Q$, $s_{saf}(x, q)$ is the confidence score of term $q$ for $x$ in the $FCL_O$ of ATGO (see Fig 1A); $y(x, q) = 1$ if $x$ is associated with $q$ in the experimental function annotation; otherwise, $y(x, q) = 0$. We minimize loss function to optimize ATGO using Adam optimization algorithm [55].

After training ATGO, the GBA strategy is used to generate the confidence score vector $s_{gba}$ of predicted GO terms for the query. Specifically, we select $K$ training proteins, which have the highest feature similarity with query, as templates, where the feature dissimilarity between two proteins is defined as the Euclidean distance of corresponding embedding vectors outputted by $FCL_a$ in trained ATGO. Then, the confidence score that the query is associated with GO term $q$ is calculated:

$$s_{gba}(q) = \frac{\sum_{k=1}^{K} w_k \cdot I_k(q)}{\sum_{k=1}^{K} w_k}, w_k = 1 - (r_k - 1)/K \tag{6}$$

where $r_k$ is the rank of the $k$-th template in $K$ templates according to the feature similarity with query; $I_k(q) = 1$, if the $k$-th template is associated with $q$ in the experimental annotation; otherwise, $I_k(q) = 0$. The values of *margin*, $c_f$, $\alpha$, and $K$ are listed in S15 Table, which are determined by maximizing the $F_{max}$ values of ATGO in the validation dataset.

## Composite model of ATGO+

Inspired by previous works [21,22], we implemented a composite model, ATGO+, by combining neural network-based model (AGTO) with sequence homology inference (SAGP), to further improve prediction accuracy:

$$s_{ATGO+}(q) = \beta \cdot s_{ATGO}(q) + (1 - \beta)s_{SAGP}(q) \tag{7}$$

where $s_{ATGO+}(q)$ is the confidence score of ATGO+ for GO term $q$, $s_{ATGO}(q)$ and $s_{SAGP}(q)$ are confidence scores for term $q$ by ATGO and SAGP, respectively. The values of the weight parameter $\beta$ are set to be 0.57, 0.60, and 0.67 for MF, BP, and CC, respectively, based on the validation dataset.

## Hierarchy of GO annotations

The GO annotation is hierarchical [33]. Specifically, for both the ground truth and the prediction, if a protein is annotated with a GO term $q$, it should be annotated with the direct parent and all ancestors of $q$. To enforce this hierarchical relation, we follow CAFA's rule and use a common post-processing procedure [9] for the confidence score of term $q$ in all GO prediction methods:

$$s(q)_{post} = \max(s(q), s(q_1^c)_{post}, s(q_2^c)_{post}, \ldots, s(q_n^c)_{post}) \qquad (8)$$

where $s(q)$ and $s(q)_{post}$ are the confidence scores of $q$ before and after post-processing, $s(q_1^c)_{post}, s(q_2^c)_{post}, \ldots, s(q_n^c)_{post}$ are the confidence scores of all direct child terms of $q$ after post-processing. This post-processing procedure enforces that the confidence score of $q$ is larger than or equal to the scores of all children.

## Evaluation metrices

$F_{max}$ and AUPR are widely used to evaluate the performance of proposed methods. $F_{max}$ is a major evaluation score in CAFA [32,33] and defined as:

$$F_{max} = \max_{0 \leq t \leq 1} \left[ \frac{2 \cdot pr(t) \cdot rc(t)}{pr(t) + rc(t)} \right] \qquad (9)$$

where $t$ is a cut-off value of confidence score; $pr(t)$ and $rc(t)$ are precision and recall, respectively, with confidence score $\geq t$:

$$\begin{cases} pr(t) = \dfrac{tp(t)}{tp(t) + fp(t)} \\ rc(t) = \dfrac{tp(t)}{tp(t) + fn(t)} \end{cases} \qquad (10)$$

where $tp(t)$ is the number of correctly predicted GO terms, $tp(t)+fp(t)$ is the number of all predicted GO terms, and $tp(t)+fn(t)$ is the number of GO terms in experimental function annotation.

AUPR is the area under the precision-recall curve and ranges from 0 to 1. In addition, AUROC is the area under the receiver operating characteristic curve and also ranges from 0 to 1.

## Supporting information

**S1 Fig. The mean AUROC values of GO terms versus 12 GO prediction methods in four ranges. (a)** range 5–10. **(b)** range 10–30. **(c)** range 30–50. **(d)** range >50.
(TIF)

**S2 Fig. The median AUROC values of GO terms versus 12 GO prediction methods in four ranges. (a)** range 5–10. **(b)** range 10–30. **(c)** range 30–50. **(d)** range >50.
(TIF)

**S3 Fig. The distributions of AUROC values for GO terms in two ranges versus 12 GO prediction methods, where the median line in the box is the median AUROC value. (a)** range 30–50. **(b)** range >50.
(TIF)

**S4 Fig. The AUPR values of 10 GO prediction methods under the sequence identity cut-off $t_1$ = 30% for three GO aspects on four individual species in CAFA3 test dataset. (a)** Human

**(b)** Arabidopsis **(c)** Fission Yeast **(d)** Mouse.
(TIF)

**S5 Fig. The architectures of different models in ablation study.**
(TIF)

**S1 Table. The *p*-values of performance difference between 12 GO prediction methods on 1068 individual test proteins under post-hoc Nemenyi test at the individual protein level, where the performance of each prediction method is measured by a group of $F_1$-scores, each of which is calculated from the predicted GO terms and native GO annotation in a single test protein.** Because the *p*-values can be only approximated in the range from 1.0e-03 to 9.0e-01 under post-hoc Nemenyi test using Python package, the numerical value of 1.0e-03 (or 9.0e-01) means that the *p*-value is below to 1.0e-03 (or upon to 9.0e-01).
(DOCX)

**S2 Table. The statistic values between SAGP and ATGO in Group A under Nemenyi post-hoc test on 1068 test proteins for MF aspect versus the increase of *K*.**
(DOCX)

**S3 Table. The prediction performance with including root GO terms for ATGO and TALE on all 1068 test proteins.** *p*-values in parenthesis are calculated between ATGO and TALE by two-sided Student's t-test. Specifically, the proposed ATGO is repeatedly implemented with 10 times on the benchmark dataset to generate the corresponding performance evaluation indices, which are compared with the fixed evaluation index generated by TALE to calculate *p*-value using two-sided Student's t-test. Bold fonts highlight the best performer in each category.
(DOCX)

**S4 Table. The summary of the proposed ATGO/ATGO+ and other 10 competing GO prediction methods on a subset of 562 test proteins which have available templates or interaction partners in all of SAGP, PPIGP, FunFams, and DIAMONDScore.** *p*-values in parenthesis are calculated between ATGO and other single-based methods and between ATGO+ and other composite methods by two-sided Student's t-test. Specifically, the proposed ATGO and ATGO+ are repeatedly implemented with 10 times on the benchmark dataset to generate the corresponding performance evaluation indices, which are compared with the fixed evaluation index generated by the competing method to calculate *p*-value using two-sided Student's t-test. Bold fonts highlight the best performer in each category.
(DOCX)

**S5 Table. The ICW-$F_{max}$ values of 12 GO prediction methods on the 1068 benchmark proteins.** Bold fonts highlight the best performer in each category.
(DOCX)

**S6 Table. The *p*-values of performance difference between 10 GO prediction methods on 3328 individual CAFA3 targets under post-hoc Nemenyi test at the individual protein level, where the performance of each prediction method is measured by a group of $F_1$-scores, each of which is calculated from the predicted GO terms and native GO annotation in a single test protein.** Because the *p*-values can be only approximated in the range from 1.0e-03 to 9.0e-01 under post-hoc Nemenyi test using Python package, the numerical value of 1.0e-03 (or 9.0e-01) means that the *p*-value is below to 1.0e-03 (or upon to 9.0e-01).
(DOCX)

**S7 Table. The ICW-$F_{max}$ values of 10 GO prediction methods on 3328 CAFA3 targets where a sequence identity cut-off $t_1$ = 30% between the training and testing proteins was applied to the five in-house methods (ATGO, ATGO+, SAGP, PPIGP and NGP).** Bold fonts highlight the best performer in each category.
(DOCX)

**S8 Table. The performance of 10 GO prediction methods under the cut-off $t_1$ = 30% on 1177 no-knowledge (NK) and 2151 limited-knowledge (LK) CAFA3 proteins.** Bold fonts highlight the best performer in each category.
(DOCX)

**S9 Table. The numbers of proteins for 20 species in CAFA3 test dataset.**
(DOCX)

**S10 Table. The prediction performance of five GO prediction methods under the cut-off $t_1$ = 100% on CAFA3 test proteins.** Bold fonts highlight the best performer in each category.
(DOCX)

**S11 Table. The prediction performance of SAGP and BLAST baseline on our constructed test dataset and CAFA3 test dataset with different cut-off values of sequence identity.** Bold fonts highlight the best performer in each category.
(DOCX)

**S12 Table. The prediction performance of ATGO models via four metric learning methods on two test datasets.** Bold fonts highlight the best performer in each category.
(DOCX)

**S13 Table. The incorrectly predicted GO terms for 12 methods on three proteins in BP aspect.**
(DOCX)

**S14 Table. The numbers of proteins and GO terms in benchmark dataset.**
(DOCX)

**S15 Table. The values of *margin*, $c_f$, $\alpha$, and *K* for three GO aspects.**
(DOCX)

**S1 Text. Sequence alignment-based GO prediction.**
(DOCX)

**S2 Text. Protein-protein interaction-based GO prediction.**
(DOCX)

**S3 Text. Naïve-based GO prediction.**
(DOCX)

**S4 Text. Friedman and Nemenyi post-hoc tests at the individual protein level.**
(DOCX)

**S5 Text. An explanation for the difference of *p*-value calculations between Student t-test and Nemenyi post-hoc test.**
(DOCX)

**S6 Text. Information content-weighted maximum $F_1$-score.**
(DOCX)

**S7 Text. Performance comparison between SAGP and BLAST baseline used in CAFA challenge.**
(DOCX)

**S8 Text. Performance comparison between four metric learning methods.**
(DOCX)

**S9 Text. The mathematics formulas for ESM-1b transformer.**
(DOCX)

**S10 Text. The functional similarity between two proteins.**
(DOCX)

## Author Contributions

**Conceptualization:** Yang Zhang.

**Data curation:** Yi-Heng Zhu.

**Funding acquisition:** Dong-Jun Yu, Yang Zhang.

**Investigation:** Yi-Heng Zhu.

**Methodology:** Yi-Heng Zhu.

**Software:** Yi-Heng Zhu.

**Supervision:** Dong-Jun Yu, Yang Zhang.

**Validation:** Yi-Heng Zhu, Chengxin Zhang.

**Writing – original draft:** Yi-Heng Zhu, Yang Zhang.

**Writing – review & editing:** Chengxin Zhang, Dong-Jun Yu.

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
