## [Decision Letter · Decision Letter 0]

7 Sep 2022

Dear Dr. Zhang,

Thank you very much for submitting your manuscript "Integrating Self-Attention Transformer with Triplet Neural Networks for Protein Gene Ontology Prediction" for consideration at PLOS Computational Biology.

As with all papers reviewed by the journal, your manuscript was reviewed by members of the editorial board and by several independent reviewers. In light of the reviews (below this email), we would like to invite the resubmission of a significantly-revised version that takes into account the reviewers' comments.

Please address the following comments from the reviewers:

Reviewer #1: please address all comments, note that the second comment suggests retraining using different embedding methods. However, that is out of scope of this work, in my opinion, so no need to do that.

Reviewer #3:

Comment 1: no need to address directly, however, if appropriate please cite the citations suggested by the reviewer.

Comment 2: Please address this comment regarding the suitability of F1 as a learning metric.

Comment 3-5: Please address

Minor comment 1: no need to address that.

We cannot make any decision about publication until we have seen the revised manuscript and your response to the reviewers' comments. Your revised manuscript is also likely to be sent to reviewers for further evaluation.

Sincerely,

Iddo Friedberg, Ph.D.

Guest Editor

PLOS Computational Biology

Nir Ben-Tal

Section Editor

PLOS Computational Biology

Please address the following comments from the reviewers:

Reviewer #1: please address all comments, note that the second comment suggests retraining using different embedding methods. However, that is out of scope of this work, in my opinion, so no need to do that.

Reviewer #3:

Comment 1: no need to address directly, however, if appropriate please cite the citations suggested by the reviewer.

Comment 2: Please address this comment regarding the suitability of F1 as a learning metric.

Comment 3-5: Please address

Minor comment 1: no need to address that.

Reviewer's Responses to Questions

**Comments to the Authors:**

Reviewer #1: This work presents a new protein function prediction approach and software. The main idea is to exploit pre-trained language models on protein sequences and use them as a basis for downstream tasks of GO term prediction. The paper is clear and well written. The architecture makes sense, the loss function for training is interesting, and the results were thorough and favor their method. Ultimately, the authors find a need to specifically combine homology-based models with network-based models.

I have some questions and concerns, mostly minor:

* The paper works off of the ESM model, however, there are other embedding systems including ProtTrans (https://arxiv.org/abs/2007.06225), TAPE (https://arxiv.org/abs/1906.08230), and Bepler & Berger's approach (https://arxiv.org/abs/1902.08661). Some mention or comments on the value of those systems would be important to make. The ProtTrans paper find their models to be superior to ESM-1b. If so, this could propagate to this paper and further improvements might be possible.

* The ablative models showed interesting performance in that M1 obtained the highest improvement in accuracy and the addition of other components only had marginal effect, particularly in MF though the trend is the same in BO and CC (Figure 5). Would other embedding methods show similar performance gain or there is something special with ESM-1b?

* Is there some interpretation in that most gain comes from the embeddings for MF. This suggest that embeddings contain some sequence signal. Have the authors tried to understand this aspect?

* The paper is written by mixing past and present tense. The authors can decide to use one. For example, "...AGTO+ achieved a further improvement overall three categories of GO terms. It also significantly outperforms..." is an example of mixing tenses. Also, "overall" -> "over all"

* life mechanism -> life mechanisms (2 places)

Reviewer #2: This manuscript reports a new deep learning method to predict the function (GO terms) of proteins. The MSA embedding techniques based on self-attend are first applied to this problem. The method performs better than both control methods and other state-of-the-art methods on the CAFA3 benchmark. Therefore, the work makes valuable contributions to the field. There is some minor issue regarding the deep learning architecture. It is not clear why three embedding components (triplet networks) are used. Some rationale should be provided to justify the design of the architecture.

Reviewer #3: The authors propose deep learning model (ATGO) for automatically assigning GO terms to protein sequences. The model uses a pre-trained language model to extract three different protein embedding vectors which are further combined into one vector using a neural network. A triplet loss is used to enforce that proteins with similar functions are close in the embedding space.

The model and a combination of the model with homology searching are compared to several other well-known function prediction methods on two datasets.

###################################################################################

Major comments

1) A main conclusion of the study is that the use of a pre-trained protein language model is by far the main source of performance for this model. However, the superiority of these language models with respect to supervised learners such as CNNs has already been extensively demonstrated (also for the CAFA3 dataset) in the past. See Rao et al. Adv Neural Inf Process Syst. 2019, Littmann et al. Scientific Reports, 2021, Villegas-Morcillo et al., Bioinformatics (2021).

2) The idea of using metric learning to learn an embedding space that reflects structural similarity is interesting (although it has also been done before for molecular function prediction for enzymes – Aman Memon et al., Bioinformatics, 2020). However, the authors here calculate the F1 score between the annotations of two proteins and pairs with F1 larger than a threshold are considered positive. But the F1 score is not a proper distance metric (e.g. doesn’t satisfy triangle inequality) and this might hamper the performance of the triplet loss. See Powers Technical report KIT-14-001 for more information on the F1 score. Perhaps a Jaccard similarity (even weighted by term information content?) might be a more suitable metric to use for metric learning.

3) The conclusion that combining ATGO with sequence homology boosts performance is not sufficiently supported. First, by looking at Tables 1 and 2, the performance improvement is in most cases <0.01 and the coverage of ATGO is already 100%. Figure 6 does show that for one protein there is a BLAST hit that helps detect some functions that were missed by the network, but looking at the numbers, this seems to be a rare occasion. So given the very small gain and that for some proteins ATGO+ can do worse than ATGO (top of Fig 6), the conclusion that information fusion leads to better performance for ATGO is not valid I think.

4) It is a bit strange that the second best method in the CAFA experiment is the blast-like baseline, although several methods that participated in CAFA3 and that were developed later (e.g. DeepGOPLus) perform considerably better than the BLAST baseline in this dataset.

5) The p-value calculations are a little problematic. First, it is not clear how the authors controlled for multiple comparisons. Additionally they don’t compare all against all, but rather ATGO against all and ATGO+ against all, but not ATGO against ATGO+ (this is related to comment 2 as well). Other comparisons such as between SAGP and PPIGP are done implicitly in the manuscript without the proper statistical test. Ideally the authors should use analysis of variants to identify whether one or multiple methods is significantly better than the rest and then use appropriate post-hoc tests to perform the pairwise comparisons.

###################################################################################

Minor comments

1) It would be nice to include an information theory-based evaluation metric such as semantic distance or IC-weighted Fmax. The number of positive examples does not always mean that the term is difficult to predict, depending on how many ‘sibling’ terms exist

2) I like the visualization of the different predictions showed in Fig 6.

**Have the authors made all data and (if applicable) computational code underlying the findings in their manuscript fully available?**

Reviewer #1: Yes

Reviewer #2: Yes

Reviewer #3: Yes

PLOS authors have the option to publish the peer review history of their article (what does this mean?). If published, this will include your full peer review and any attached files.

Reviewer #1: No

Reviewer #2: No

Reviewer #3: No
---

## [Decision Letter · Decision Letter 1]

5 Dec 2022

Dear Dr. Zhang,

We are pleased to inform you that your manuscript 'Integrating Unsupervised Language Model with Triplet Neural Networks for Protein Gene Ontology Prediction' has been provisionally accepted for publication in PLOS Computational Biology.

Before your manuscript can be formally accepted you will need to complete some formatting changes, which you will receive in a follow up email. A member of our team will be in touch with a set of requests. At this point, please also pay special attention to the typographical errors as pointed out by the reviewers.

Best regards,

Iddo Friedberg, Ph.D.

Guest Editor

PLOS Computational Biology

Nir Ben-Tal

Section Editor

PLOS Computational Biology

Reviewer's Responses to Questions

**Comments to the Authors:**

Reviewer #1: The authors have responded well to the criticism. I notice a few more lingering typos and the authors could use my suggestions below to further clean up the paper.

Summary:

"all gene and gene products" -> "all genes and gene products"

Introduction:

"we utilized a unsupervised" -> "we utilized an unsupervised"

Results:

"Fmax" -> max should be in the subscript (potentially also 1 for F1)

"we perform Friedman test" and then "Thus, a Nemenyi post-hoc test" shortly after. As far as I understand, this should be the same test, sometimes referred to as Friedman-Nemenyi.

"did not remove homologous" -> "did not remove homologs"

"designed as following" -> "designed as follows"

"with Sigmoid activation function" -> "with sigmoid activation function"

Reviewer #2: My comments have been addressed.

Reviewer #3: The authors have addressed all my comments.

Small typo at the top of page 6, 'analysis of variants' should be 'analysis of variance'

**Have the authors made all data and (if applicable) computational code underlying the findings in their manuscript fully available?**

Reviewer #1: Yes

Reviewer #2: Yes

Reviewer #3: Yes

PLOS authors have the option to publish the peer review history of their article (what does this mean?). If published, this will include your full peer review and any attached files.

Reviewer #1: No

Reviewer #2: No

Reviewer #3: No

---

## [Editor Report · Acceptance letter]

16 Dec 2022

PCOMPBIOL-D-22-01039R1 

Integrating Unsupervised Language Model with Triplet Neural Networks for Protein Gene Ontology Prediction

Dear Dr Zhang,

I am pleased to inform you that your manuscript has been formally accepted for publication in PLOS Computational Biology. Your manuscript is now with our production department and you will be notified of the publication date in due course.

With kind regards,

Anita Estes
